# Neuraminidase Inhibitor Zanamivir Ameliorates Collagen-Induced Arthritis

**DOI:** 10.3390/ijms22031428

**Published:** 2021-01-31

**Authors:** Bettina Sehnert, Juliane Mietz, Rita Rzepka, Stefanie Buchholz, Andrea Maul-Pavicic, Sandra Schaffer, Falk Nimmerjahn, Reinhard E. Voll

**Affiliations:** 1Department of Rheumatology and Clinical Immunology, Medical Center–University of Freiburg, Faculty of Medicine, University of Freiburg, 79106 Freiburg, Germany; mietz@immunology.uzh.ch (J.M.); Rita.Rzepka@uniklinik-freiburg.de (R.R.); buchholz.st@gmx.de (S.B.); Andrea.Maul-Pavicic@uniklinik-freiburg.de (A.M.-P.); Sandra.Schaffer@uniklinik-freiburg.de (S.S.); 2Department of Biology, Institute of Genetics, Friedrich-Alexander University Erlangen-Nürnberg (FAU), 91058 Erlangen, Germany; falk.nimmerjahn@fau.de

**Keywords:** arthritis, mouse model, sialylation, neuraminidase, neuraminidase inhibitor

## Abstract

Altered sialylation patterns play a role in chronic autoimmune diseases such as rheumatoid arthritis (RA). Recent studies have shown the pro-inflammatory activities of immunoglobulins (Igs) with desialylated sugar moieties. The role of neuraminidases (NEUs), enzymes which are responsible for the cleavage of terminal sialic acids (SA) from sialoglycoconjugates, is not fully understood in RA. We investigated the impact of zanamivir, an inhibitor of the influenza virus neuraminidase, and mammalian NEU2/3 on clinical outcomes in experimental arthritides studies. The severity of arthritis was monitored and IgG titers were measured by ELISA. (2,6)-linked SA was determined on IgG by ELISA and on cell surfaces by flow cytometry. Zanamivir at a dose of 100 mg/kg (zana-100) significantly ameliorated collagen-induced arthritis (CIA), whereas zana-100 was ineffective in serum transfer-induced arthritis. Systemic zana-100 treatment reduced the number of splenic CD138^+^/TACI^+^ plasma cells and CD19^+^ B cells, which was associated with lower IgG levels and an increased sialylation status of IgG compared to controls. Our data reveal the contribution of NEU2/3 in CIA. Zanamivir down-modulated the T and B cell-dependent humoral immune response and induced an anti-inflammatory milieu by inhibiting sialic acid degradation. We suggest that neuraminidases might represent a promising therapeutic target for RA and possibly also for other antibody-mediated autoimmune diseases.

## 1. Introduction

Rheumatoid arthritis (RA) is a systemic autoimmune disease which is characterized by autoimmune-driven joint inflammation, leading to cartilage and bone destruction. The origin of RA is largely unknown. In recent years, research has revealed that genetic predisposition, bacterial and viral triggers, as well as environmental risk factors contribute to the development of RA [1]. The development of therapies for RA to control inflammatory arthritis has increased rapidly [2,3]. The exploration of molecular targets contributing to RA pathogenesis has led to the development of so called biologics and small-molecule inhibitors. Targeted drugs such as tumor necrosis factor alpha antagonists (TNF-alpha) (e.g., infliximab) and interleukin-6 receptor (IL-6R) antagonist (e.g., tocilizumab), as well as inhibitors of janus kinases (JAKs) (e.g., tofacitinib), have markedly advanced the treatment of RA [3,4,5]. Although in the majority of patients they are well tolerated, adverse effects such as infections, nausea, vomiting, diarrhea, liver problems, injection site reactions, etc., are associated with biologics. Moreover, 30–40% of RA-patients insufficiently respond to biologics [6,7]. In order to improve the treatment and outcome of RA and to reduce severe adverse effects, this research aims to unravel further mechanisms that drive inflammation in RA and may represent targets for therapeutic interventions.

The development of pathogenic autoantibodies correlates with inflammatory activity in RA and in experimental arthritis models [8,9]. A number of studies have shown the relevance of rheumatoid factor, anti-citrullinated protein antibodies (ACPAs), and anti-carbamylated protein antibodies and also emphasized the role of the aberrant N-glycosylation of immunoglobulin G (IgG) in RA pathogenesis [9]. IgG effector immune functions are not only dependent on the IgG subclasses but is also on IgG galactosylation and sialylation [10,11,12,13]. In progressive RA, IgG is less galactosylated and sialylated [14,15,16]. Anthony et al. reported that the anti-inflammatory activity of intravenous immunoglobulin G (IVIG) is dependent on sialylated fractions [17]. In murine immune thrombocytopenia, the sialylation-independent actions of IVIG also exert anti-inflammatory properties [18]. Nevertheless, altering IgG sialylation status might represent an immunomodulatory treatment approach for antibody-mediated autoimmune diseases such as RA. In contrast, the impact of the dynamics of cell surface sialylation in RA pathogenesis is not fully understood. In the late 1980s, Gorczyca et al. described that the degree of sialylation of macrophage surface glycoconjugates modulates IgG-Fc-gammaR [19]. Furthermore, sialylated N-glycans on the cell surface contribute to the suppression of the phagocytotic capability of mouse monocytic cells [20]. Moreover, it was shown that the alpha-(2,6)-sialylation levels were downregulated under inflammatory conditions during the conversion from immature to mature dendritic cells [21].

The sialylation of proteins and lipids is regulated by sialyltransferases (STs) and neuraminidases (NEUs), also called sialidases. STs are enzymes that transfer sialic acid (SA) to de novo synthesized oligosaccharides by a complex biosynthesis pathway. N-acetylneuraminic acid (Neu5Ac) is the most abundant form of an SA and can be linked either through an alpha-(2,3)- or an alpha-(2,6)-linkage to subterminal galactose by specific STs—namely, beta-galactoside alpha-(2,3)-sialyltransferase (ST3GAL) or β-galactoside alpha-(2,6)-Sialyltransferase ST6GAL [22].

The removal of SA can be catalyzed by NEUs, a family of exoglycosidases (EC 3.2.1.18). NEUs are able to cleave terminal SA residues from mono- and oligosaccharides of glycoproteins- and lipids and exist in bacteria, viruses, protozoans, and mammales. In humans, four types (NEU1, NEU2, NEU3, and NEU4) are described. Human NEUs (hNEUs) differ in their substrate specificity and intracellular location. NEU1 is predominantly located in the lysosomal compartment and plasma membrane after activation, whereas NEU2 can be found in the cytosolic compartment. NEU3 is mainly located in the plasma membrane and NEU4 exerts its activity in lysosomes, mitochondria, and the endoplasmic reticulum (ER). The endogenous expression of neuraminidases is tissue- and species-specific. In human tissues, NEU1 shows the strongest expression, which is 10–20 times greater than NEU3 and NEU4 expression. NEU2 is expressed at extremely low levels in a range of tissues. NEU expression in tissues of mice and rats has not been fully explored [23,24].

Agents that potently inhibit viral NEUs (vNEUs) [25] of the influenza virus are oseltamivir (Tamiflu^®^) and zanamivir (Relenza^®^), which are both approved for the treatment of influenza infection [26,27]. Although these drugs inhibit vNEUs in a low nanomolar range, these agents have limited inhibitory efficacy for human NEUs. In the study of Hata et al. [22], oseltamivir hardly affected the activities of the four human NEUs, even at a concentration of 1 mM, while zanamivir significantly inhibited the human sialidases NEU3 and NEU2 in the micromolar range (K*i*, 3.7 µM ±0.48 and 12.9 µM ±0.07 µM, respectively) [23,24].

To the best of our knowledge, studies that elucidate the role of NEUs in arthritis are limited [28]. Hence, we aimed to investigate the role of NEUs in arthritis and analyze the potential therapeutic effect of neuraminidase inhibition on collagen-induced arthritis (CIA) and on serum-induced arthritis (STIA) because a critical role of asialylated Igs in RA is likely.

## 2. Results

### 2.1. Role of NEU2/3 Inhibition in CIA and STIA

#### 2.1.1. Zanamivir Treatment Ameliorates Collagen-Induced Arthritis But Not Serum Transfer-Induced Arthritis

CIA, a commonly used model of RA, can be induced in susceptible mouse strains [29,30] with a single intradermal injection of CII emulsified in CFA [31,32]. In CIA, a broad range of immune cells contribute to progressive joint inflammation, leading to cartilage and bone destruction. The involvement of T and B cells triggers the immunization phase, leading to the production of complement-fixing anti-CII autoantibodies and the infiltration of lymphocytes, monocytes, and neutrophils into the affected joints. The release of pro-inflammatory cytokines such as TNF-alpha, interleukin-1, interleukin-6, and interleukin-17 from activated cells amplifies inflammatory arthritis [33,34].

First, we examined the effect of zanamivir, a sialic acid-analogue neuraminidase inhibitor [35], on the severity of arthritis. The experimental schedule is presented in Figure 1a. CII-immunized mice received daily intraperitoneal injections of zanamivir at a dose of 50 mg/kg (zana-50) or 100 mg/kg (zana-100) or vehicle (control group). Treatment started two days before CII immunization to ensure the effective inhibition of neuraminidase activity at the time of immunization. As presented in Figure 1b, zanamivir reduced arthritis severity in a dose-dependent manner. Zana-100 treatment reduced arthritis significantly from day 31 to day 41 compared to the control group. The significant treatment effect of zanamivir at a dose of 100 mg/kg is further shown by the area under the curve (AUC) of the score (Figure 1c). Moreover, in CIA disease onset was delayed in mice receiving 100 mg/kg of zanamivir. In the vehicle and zana-50 group, the first signs of arthritis were observed at day 25 post CII immunization, whereas in the high-dose group joint swelling started to develop at day 31 (Figure 1b). The prevalence of arthritis is depicted in Figure 1d. Compared with controls significantly lower numbers of arthritic mice were observed under zanamivir until day 36. However, on day 38 approximately 70% of the mice developed arthritis (score >0) (Figure 1d).

Second, we investigated the potential therapeutic effect of zanamivir in STIA. STIA reflects the immune complex-mediated phase of arthritis and is T and B cell-independent, because arthritis-inducing antibodies are injected. A key role in the pathogenesis of STIA is played by monocytes/macrophages, neutrophils, and mast cells (Figure 1e) [36]. Here, the experiment clearly showed that high systemic doses of zanamivir administered over a period of 9 days did not exert beneficial effects on STIA (Figure 1f). Similarly, the viral neuraminidase inhibitor oseltamivir, which has also some inhibitory activity for human NEU1 and alters leukocyte extravasation, did not improve the clinical signs of STIA at a rather high dose of 100 mg/kg (data not shown).

Moreover, the welfare of mice was monitored daily during the experiments. Parameters of general health (body weight, mobility, appearance, posture, vocalization, dehydration, feces) (data not shown) were not altered in zanamivir-treated mice compared to the vehicle. Even at the rather high dose of 100 mg/kg zanamivir, the substance was well-tolerated without evidence of toxicity.

In conclusion, zanamivir at a dose of 100 mg/kg significantly ameliorated CIA, an experimental arthritis model that depends on B and T cells during the immunization phase. In contrast, zanamivir did not improve the effector phase of arthritis, which is resembled in the model of STIA, suggesting that the sialylation pattern of the applied arthritogenic serum is not markedly modified by zanamivir. Moreover, the previously described inhibitory effects of oseltamivir on cell extravasation did not result in an improvement of STIA.

#### 2.1.2. Zanamivir Inhibits Production of Anti-CII Autoantibodies and Total IgG in CIA

A hallmark of CIA is the production of pathogenic anti-CII autoantibodies. Antibody deposition to cartilage, immune complex formation followed by the activation of complement factors, and binding to Fc receptors (FcRs) play important roles in the induction and perpetuation of the clinical disease. Hence, we analyzed the production of anti-CII IgG antibodies and anti-CII IgG subclasses IgG1, IgG2a, and IgG2b at the indicated time points post CII-immunization.

IgG antibodies against bovine CII and all subclasses (IgG1, IgG2a, and IgG2b) were detected as expected 19 days after arthritis induction. In vehicle and zana-50-treated mice the levels of anti-CII IgG as well as IgG2a and IgG2b significantly increased from day 19 to day 41, whereas mice treated with 100 mg/kg of zanamivir showed no significant elevation in anti-CII IgG, IgG2a, and IgG2b levels between day 19 and day 41. On day 41, there was a numeric reduction in the anti-CII IgG, IgG2a, IgG2b and IgG1 levels in the zana-100 group vs. vehicle group, but this difference did not reach statistical significance due to the high standard deviations. Nevertheless, the data indicate an inhibition of anti-CII autoantibody production by zanamivir at a dose of 100 mg/kg in the late phase of CIA (Figure 2a–d). The ratio of anti-CII IgG2a to anti-CII IgG1 was not altered indicating that zanamivir did not induce a switch towards a TH2-mediated response (Figure 2e).

Further, total IgG increased significantly over the observation period in the vehicle and zana-50 treatment group. An inhibition of IgG production was measured in the zana-100 group between day 19 and day 41. On day 41, the total IgG concentrations were significantly reduced after zana-50 and zana-100 treatment compared to the controls (Figure 2f).

Together, the data suggest that high-dose zanamivir inhibits the humoral immune response in CII-immunized mice. Thereby the formation of immune-complexes might be diminished, resulting in decreased inflammation.

### 2.2. Zanamivir Treatment Decreases Absolute Numbers of CD138^+^/TACI^+^ Plasma Cells and CD19^+^ B Cells in CIA

A complex network of cells such as T cells, B lineage cells, and myeloid cells is involved in the pathogenesis of RA. Recent data demonstrated that B cells and autoantibody-secreting plasma cells might be potential targets not only for the treatment of systemic lupus erythematosus (SLE) and multiple sclerosis (MS), but also for RA [37]. We investigated the effect of zanamivir on numbers of splenic T cells, CD19^+^ B cells, CD138^+^/TACI^+^ plasma cells (PCs), and CD11b ^+^ myeloid cells in CIA. Overall, a dose of 100 mg/kg of zanamivir lowered the splenic cell numbers significantly compared to the control group (Figure 3a). Absolute numbers of CD4^+^CD8^−^-T cells, CD4^-^CD8^+^-T cells, and also CD11b^+^cells were not altered between the groups (Figure 3b–d). However, when we analyzed the numbers of CD138^+^/TACI^+^ plasma cells and CD19^+^ B cells, we observed significantly reduced numbers after zana-100 treatment compared to the controls (Figure 3e,f).

In conclusion, zanamivir decreases B cells and autoantibody-secreting plasma cells in CIA. This effect may explain the decreased anti-CII antibody concentrations and the observed anti-arthritic effect of zanamivir.

### 2.3. Role of NEU2/3 Inhibition on Sialylation Pattern in CIA

#### 2.3.1. Zanamivir Increases Sialic Acid Content of Circulating IgG During CIA

IgG sialylation plays an important role in modulating the inflammatory activity of antibodies [11]. The levels of (2,6)-SA residues on total serum IgG were measured by ELISA using biotinylated Sambucus Nigra agglutinin lectin (SNA), which specifically binds to terminal-linked (2,6)-SA. The extent of IgG sialylation is expressed as ratio of SNA to total IgG. As expected, in vehicle-treated mice the SNA:IgG ratio decreased over time during arthritis development (between day 19 and 41), which is in accordance to other studies showing the inflammatory properties of desialylated IgG and a decrease in IgG sialylation during inflammatory diseases [16,38]. Significantly reduced IgG sialylation was also detected in zana-50 mice, although it was numerically less than in the placebo group. In contrast to vehicle- and zana-50-treated mice, in zana-100 treated mice the SNA:IgG ratio did not significantly decrease between day 19 and 41. The difference in the ratios of SNA binding:total IgG between the vehicle and zana-100 group at day 41 just missed statistical significance (Figure 4, day 41).

Thus, the desialylation of IgG—probably extracellular—appears to be blocked by Neu2/3 inhibition, resulting in antibodies with anti-inflammatory properties. Currently, we cannot clearly distinguish if zanamivir inhibits antibody desialylation primarily during antibody synthesis/secretion or in the extracellular space.

#### 2.3.2. Zanamivir Treatment Leads to Increased Levels of (2,6)-sialic Acid Content on the Cell Surface of CD138^+^/TACI^+^ Plasma Cells

The biological function of sialic acids on the surfaces of cells and their contribution to the immunopathogenesis of arthritis remains elusive. We analyzed the content of 2,6-linked sialic acid residues on the surface of CD138^+^/TACI^+^ plasma cells, CD19^+^ B cells, CD4^+^CD8^−^-T cells, and CD11b^+^ cells of CII immunized mice under zanamivir treatment. Splenocytes were stained for the respective surface molecules and counterstained with *SNA*, a lectin that binds preferentially to sialic acids linked to terminal galactose in alpha-(2,6) and, to a much lesser degree, alpha-(2,3) linkage, and analyzed them by flow cytometry. The sialic acid level is expressed as the mean fluorescence intensity (MFI). Flow cytometric analysis showed no changes in the SA levels on CD4^+^CD8^−^T cells, CD11b^+^ cells, or CD19^+^ B cells of zana-100 treated mice relative to the controls (Figure 5a,b,d). However, significantly higher levels of SA were observed on CD138^+^/TACI^+^ PCs after zana-100 treatment compared to vehicle and zana-50-treated mice (Figure 5c).

In summary, in CIA NEU2/3 inhibition by zanamivir increases selectively the sialylation of antibody-secreting cells.

### 2.4. Role of NEU 2/3 Inhibition on Neuramidase Activity in CIA

#### Zanamivir-Treated Arthritic Mice Show Reduced Neuraminidase Activity in Arthritic Joints

Next, a fluorescence-based assay was used to assess the neuraminidase activity in joint and spleen extracts of CIA mice. The assay is based on the conversion of a fluorogenic substrate (2′-(4-methylumbelliferyl)-α-D-N-acetylneuraminic acid (MUNANA)) into α-D-N-neuraminic acid and the fluorescent molecule 4-methylumbelliferone in the presence of neuraminidase (Figure 6a) [39]. Tissue extracts of hind paws and spleens were prepared immediately after the euthanization of the animals and applied to the assay. Neuraminidase activity in samples was calculated as µU/mg total protein. The joint extracts of vehicle-treated mice showed significantly higher neuraminidase activity compared to the joint extracts of zana-50 and zana-100-treated mice (Figure 6b). Overall, the joint extracts of CIA mice had a more than 10-fold higher neuraminidase activity than spleen extracts. Further, in spleens no significant differences were observed between the groups (Figure 6c).

In conclusion, an overall higher level of NEU activity was found at the site of inflammation and can be selectively inhibited by a NEU 2/3 inhibitor.

## 3. Discussion

RA is a chronic inflammatory joint disease that develops over various critical stages from healthy to established RA [1]. In the last few decades, biologics such as TNF- and IL-6 antagonists, and recently JAK inhibitors have markedly improved the outcome of RA treatment. However, even today a substantial number of patients do not sufficiently respond to treatment and do not reach remission or low disease activity [3,4,5]. Hence, continued efforts have to be undertaken to explore new cellular and molecular targets to develop additional options for RA therapy. An important factor that might critically contribute to RA pathogenesis is the aberrant sialylation profile of autoantibodies, which is orchestrated by STs and NEUs. However, despite its known biological relevance in oncogenesis, reprogramming, embryonic development, and immune responses [40], the role of NEUs in autoimmune diseases is incompletely understood.

To the best of our knowledge, this is the first study so far demonstrating a treatment effect of neuraminidase inhibition in arthritis. We have shown that neuraminidase inhibition by zanamivir, an inhibitor of mammalian NEU2 and NEU3, significantly ameliorates the clinical symptoms of CIA and delays disease onset. In CIA, zanamivir reduced the numbers of B-lineage cells, which was accompanied by an inhibition in the humoral immune response. Furthermore, at sites of inflammation the NEU activity was significantly lower after zanamivir treatment, proving its mode of action.

We show that zanamivir at 100 mg/kg prevented an increase in anti-CII IgG antibody levels and in total IgG concentrations during the arthritic phase of CIA. Instead of an increase in antibody levels during arthritis development in vehicle and zana-50 groups, the antibody levels did not further rise during the CIA effector phase in high-dose zanamivir-treated mice, a fact that is likely to cause the amelioration of arthritis. In CIA, arthritis can be induced by a single intradermal injection of CII emulsified in CFA [32]. This model strongly depends on the activation of auto-reactive T and B cells [41] leading to the production of anti-CII autoantibodies that bind to cartilage structures. The formation of local immune complexes evokes complement fixation and finally the attraction of neutrophils, monocytes, T cells, and B cells into the joint space, which resembles the effector phase of arthritis. Not only in RA but also in CIA a key role in pathogenesis is attributed to B cells [42]. Mice lacking antibody production due to genetically modified B cells are protected against CIA [43]. Additionally, the depletion of B cells by anti-CD20 antibodies prevents CIA [44]. The significant role of germinal center (GC) formation in CIA was shown by Dahdah et al. Anti-CII antibody responses were strongly dependent on formation of GCs [45]. Additionally, other B cell compartments in secondary lymphoid tissues influence the development of CIA. Carnot et al. showed that naive DBA/1 mice naturally display CII-reactive marginal zone B cells producing anti-CII autoantibodies prior to immunization, which expand after CII immunization [46]. Our data suggest that the zanamivir-induced delay of CIA onset might be a result of an altered GC reaction and/or of a modulation of other B cell compartments. Our observation of reduced B-lineage cell numbers including antibody secreting plasma cells further underscores this assumption. One limitation of our study is that we did not directly investigate the formation of GCs or the role of marginal zone (MZ) and follicular (FO) B cells under zanamivir in CIA, which was beyond the scope of this study. Given this limitation, further research on lymphoid or local structures is needed to investigate the underlying mechanisms of the observed therapeutic effect of NEU2/NEU3 inhibition in CIA.

The effector phase of arthritis can be mimicked by STIA. Here, arthritogenic antibodies contained within K/BxN serum bind to ubiquitously expressed glucose-6-phosphate isomerase (GPI) and provoke arthritis independently of T and B cell responses [47]. In contrast to CIA, zanamivir failed to mitigate arthritis in STIA. Recently, it was shown that the mobilization of neutrophil sialidase activity desialylates the pulmonary vascular endothelial surface and increases resting neutrophil adhesion to and migration across the endothelium. Traversing the endothelial cell layers in vitro was inhibited when activated PMNs were treated with 2,3-dehydro-2-deoxy-N-acetylneuraminic acid (DANA) or an anti-neuraminidase antibody. Moreover, the authors demonstrated that the inhibition of endothelial neuraminidases does not impair the adhesion and transmigration of PMNs [48]. A study by Feng et al. demonstrated that the stimulation of PMNs leads to the translocation of NEU1 from the cytosol to the cell surface. Subsequently, an epitope on Mac-1 (CD11b/CD18) is exposed and enhances the interaction with ICAM-1 on endothelial cells [49]. However, cell–cell or cell–matrix interactions are not only regulated by NEU activity. The complex sialic acid metabolism in diapedesis is also orchestrated by the activity of STs. Recently, it was shown that ST activity is required for optimal PMN migration in vitro and in vivo in response to IL-8 [50]. Because neuraminidase inhibitors and STs may act on different molecules, including cell surface receptors, and influence their function, they could exert different or even opposing effects. We observed in STIA that neither zanamivir treatment nor oseltamivir (data not shown), an inhibitor of NEU1, ameliorated arthritis, although this model depends on the extravasation of inflammatory cells such as neutrophils through the endothelium into joints. The failure to ameliorate STIA by NEU inhibition led us to the assumption that NEUs are not crucially involved in the effector phase of arthritis. However, the role of neutrophilic NEU1 in STIA is still open and would require further analysis. However, our observations strengthen the suggestion that zanamivir-induced inhibition of NEU2/NEU3 modulate predominantly the T and B cell-dependent onset phase of CIA, thereby inhibiting the humoral inflammatory response.

Recent reports have discussed the fact that the sialylation status of antibodies determines pro- and anti-inflammatory activities [11,16,51,52]. Ohmi et al. highlighted the role of hyposialylated ACPA-IgGs and anti-CII IgGs in human RA and in mouse models of arthritis. Mice with a conditional ST6Gal1 deficiency in activated B cells showed an exacerbation of joint inflammation after CII immunization. In contrast, artificial sialylation of anti-CII antibodies, including ACPAs, attenuated collagen-antibody induced arthritis (CAIA). These experiments demonstrated that sialylation regulates the pathogenicity of RA-associated IgGs, presenting a promising target for antigen-specific immunotherapy [16]. During arthritis development, we observed decreasing levels of (2,6)-linked SA on IgG molecules in the control group, whereby zanamivir-treated CIA mice have a less pronounced desialylation on IgG molecules, indicating the contribution of NEU2/3 to IgG hyposialylation during arthritis.

Despite the knowledge of antibody sialylation in RA, little is known about the relevance of cell surface sialylation in autoimmunity. Lee and Wang reviewed the aberrant sialylation of immune cells in RA [40] and mainly discussed Liou’s [28] findings. In 2016, Liou et al. found that the ST3Gal-1 and Neu3 levels in blood B cells correlated positively with moderate and high disease activity DAS28 scores in RA patients [28]. However, in this study the authors did not characterize the impact of (2,6)-linked SA on B cells or other cell types in RA. Our FACS data revealed that the (2,6)-linked SA levels were significantly higher on CD138^+^/TACI^+^ plasma cells upon zanamivir treatment. No alterations were observed on cell surfaces of T cells and CD11b^+^ myeloid subsets. These findings imply that NEU2 and/or NEU3 are involved in regulating the sialylation pattern in plasma cells. However, the functional relevance of hypersialylation in B and plasma cells for the amelioration of clinical manifestation in experimental arthritis is not yet fully clarified. Possibly, the inhibition of NEUs leads to the increased sialylation of secreted antibodies as well as surface proteins of plasma cells.

The role of endogenous mammalian neuraminidases in autoimmunity and the potential effects of primarily viral neuraminidase inhibitors such as zanamivir are not yet well characterized. This is due to the fact that neuraminidase inhibitors such as oseltamivir or zanamivir display a rather low affinity to mammalian sialidases in vitro and in vivo. At least zanamivir is able to inhibit NEU3 and NEU2 at micromolar concentrations, whereas oseltamivir shows hardly any effect against mammalian sialidases even at 1 mM in contrast to their therapeutic activity as anti-viral drug [24]. Additionally, DANA, a sialic acid analogue, which blocks all NEUs, has a more than ten times lower blocking activity against mammalian NEUs compared to viral and bacterial NEUs. Main functions, in particular of NEU1, are described for cancer progression [53]. The immunoregulatory activities of NEU1 were described by Nan and co-workers [54]. Apart from degrading glycoconjugates in lysosomes, NEU1 is also involved in cellular signaling in immune responses. NEU activity is increased in T lymphocytes activated by anti-CD3 and anti-CD28 antibodies and contributes to the hyposialylation of certain surface glycoconjugates and to the production of interferon-γ (IFN-γ) [54]. Upon T cell receptor stimulation, NEU1 and NEU3 mRNAs are significantly increased and induce several cytokines, including IL-2 and IL-13 [55]. During the differentiation of monocytes to macrophages, neuraminidases are differentially expressed and, in contrast to NEU2 and NEU4, the amounts of NEU1 and NEU3 increase [56]. These data show that NEU expression depends on the activation state of the cell. In CIA, we detected an overall higher NEU activity at sites of inflammation compared to secondary lymphoid tissues. Further, we demonstrated that NEU activity was inhibited by zanamivir at sites of inflammation in contrast to the activity in the spleen. Based on the observed beneficial effect of zanamivir in CIA, we conclude that NEU2/3 inhibition may create a local anti-inflammatory milieu.

In summary, the present study identified the critical role of NEU2 and/or NEU3 in CIA. Zanamivir ameliorates CIA by a B cell-suppressing effect. Further, NEU2/3 inhibition may cause anti-inflammatory properties by preventing the degradation of alpha-(2,6)-linked sialic acid residues on IgGs and PC surface proteins, most likely during their synthesis. Future work should investigate the differential roles of NEU2 and NEU3. To this end NEU inhibitors with high selectivity for individual mammalian NEUs will be an important tool to study the contribution of NEUs in more detail and their potential use in the treatment of RA and other antibody-mediated autoimmune diseases. Our data set the stage for the investigation of NEU inhibition as a novel treatment target in immune-mediated joint inflammation.

## 4. Materials and Methods

### 4.1. Mouse Models of Arthritis and Scoring

Animal experiments were approved by the local governmental commission for animal protection of Freiburg (Regierungspräsidium Freiburg, approval no. G17/10). Mice were maintained in conventional housing with 5 mice per cage under controlled 12 h light/12 h dark cycles. Collagen-induced arthritis (CIA) was induced in DBA/1J mice (Janvier Laboratories, Le Genest-Saint-Isle, France) at the age of 9 weeks. At day 0 mice were immunized with a single intradermal injection of bovine collagen type II (CII) (Chondrex, Inc., Redmond, WA, USA) emulsified in complete Freund’s adjuvant (CFA) (Chondrex, Inc., Redmond, WA, USA) [31]. Serum transfer-induced arthritis (STIA) was performed in 9 week old C57BL/6J mice (Janvier Laboratories, Le Genest-Saint-Isle, France) [57]. Mice received a single intraperitoneal injection of K/BxN serum (kindly provided by Falk Nimmerjahn, Erlangen, Germany) at day 0. Mice were inspected macroscopically for clinical signs of arthritis by 2 independent observers under blinded conditions. Arthritis was graded on a scale of 0–4: 0 = no swelling; 1 = swelling of one joint; 2= moderate swelling of > one joint; 3 = extensive swelling > one joint. 4 = severe swelling of the entire paw. Each paw was individually graded and scores were summed yielding a maximum score of 16 per mouse. Mean arthritis score (MAS) is expressed as mean of scoring points per group. The area under the curve (AUC) of the arthritis score was calculated from day 20 to day 41. The percentage (%) of symptomatic mice (score >0) was indicated prevalence of arthritis.

### 4.2. Treatment of CIA and STIA with Zanamivir

The neuraminidase inhibitor zanamivir (Relenza^®^; GlaxoSmithKline GmbH & Co. KG, München, Germany) was dissolved in PBS (Sigma-Aldrich, Taufkirchen, Germany). Relenza is provided as a powder supplemented with lactose monohydrate (Sigma-Aldrich, Taufkirchen, Germany). Therefore, vehicle-treated mice received a solution of PBS supplemented with lactose monohydrate. Treatment was performed once daily and was started 2 days before CII immunization or serum transfer. Mice were divided into three groups. Group A received the vehicle. Group B received 50 mg/kg of zanamivir (low dose group). Group C received 100 mg/kg zanamivir (high dose group). Vehicle: *n*= 16; 50 mg/kg zanamivir: *n* = 17; 100 mg/kg zanamivir *n* = 7.

### 4.3. Determination Measurement of anti-CII Autoantibodies and Total IgG

Microtiterplates (Nunc Maxsorp, ThermoFisher Scientific, Waltham, MA, USA) were coated overnight at 4 °C with 100 µL of 10 µg/mL CII in PBS. 2% (wt/vol) BSA (Carl Roth, Karlsruhe, Germany) in PBS was used to prevent non-specific binding. To detect anti-CII IgG and anti-CII IgG1 sera were diluted 1:1000 in PBS. For the determination of anti-CII IgG2a levels and anti-CII IgG2b levels sera were diluted 1:4000. After an incubation period of 90 min at room temperature isotypes of captured anti-CII IgG antibodies were detected using respective horseradish peroxidase (HRP)-labeled goat anti mouse IgG antibodies (SouthernBiotech, Birmingham, AL, USA) diluted to 1:4000 in 1% (wt/vol) BSA/PBS. The detection antibodies were incubated for 1 h. After washing ABTS substrate solution (Sigma-Aldrich, Taufkirchen, Germany) was added. Absorbance was measured at 405/490 by a MultiskanTM FC Microplate Photometer (Thermo Fisher Scientific, Waltham, MA, USA). ELISA was performed in duplicates. Titers of CII-specific autoantibody titers were expressed as relative units/mL. Therefore, a high titer standard was generated using pooled sera of CIA mice. The high titer standard was set as 100 U/mL and unknowns were calculated using a nonlinear regression standard curve.

Measurements of total IgG were performed on microtiter plates coated with goat anti-mouse IgG (SouthernBiotech, Birmingham, AL, USA) diluted to 2 µg/mL in 0.1 M carbonate-bicarbonate buffer pH 9.5. After blocking with 2% (wt/vol) BSA in PBS for 1 h at room temperature, serum dilutions prepared in PBS/2% FCS were added to the plate. Mouse IgG standard (Dianova, Hamburg, Germany) served as standard and a dilution series was performed from 100 ng/mL to 0.8 ng/mL. Sera and standard were incubated for 2 h at room temperature. Bound serum IgG was detected with HRP-labeled goat anti-mouse IgG (Dianova, Hamburg, Germany at 0.8 mg/mL) diluted to 1:10,000 in PBS/2% FCS. After washing, ABTS solution was added and absorbance at 405/490 nm was measured by a Microplate Photometer. ELISA was performed in duplicates. The amount of total serum IgG was interpolated from the mouse IgG standard curve using nonlinear regression curve and expressed as total IgG (mg/mL).

### 4.4. Determination of Alpha-Linked 2,6-sialic Acids on Total IgG by ELISA

Alpha-2,6-linked sialic acid on IgG was detected by Sambucus Nigra agglutinin (SNA)-ELISA. Coating was performed using 1 µg/mL of F(ab’)_2_ fragment goat anti mouse IgG (γ-chain specific) (Jackson ImmunoReseach, West Grove, PA, USA) on microtiter plates at 4 °C. Free binding sites were blocked using 1× CarboFree Blocking Solution (Vector Laboratories, Burlingame, CA, USA). Sera were diluted to 1:2000 in PBS and incubated for 1 h at room temperature. Afterwards, biotinylated SNA (Vector Laboratories) was diluted to 1 µg/mL in lectin buffer (PBS, 0.1% Tween-20, 1 mM CaCl2 1 mM MgCl2) and added to the wells for a 1 h. Subsequently, HRP-conjugated streptavidin (BioLegend, Koblenz, Germany) was diluted to 1:1000 in LowCrossBuffer^®^ (Candor, Wangen, Germany) and incubated for 1 h at room temperature. ABTS solution was added and absorbance at 405/490 nm was measured. ELISA was performed in duplicates. SNA levels were normalized to the total IgG content and expressed as a ratio of SNA:IgG

### 4.5. Phenotyping of Splenocytes and Determination of 2,6-sialytated Cell Surface Subsets by Flow Cytometry

Flow cytometric analysis was performed on splenic single cell suspensions using fluorochrome-conjugated monoclonal antibodies. A total of 5 *×* 10^6^ splenocytes were stained with 25 ng SNA-FITC solution (Vector Laboratories, Burlingame, CA, USA) in a volume of 100 µL for 20 min at 4 °C. After washing with ice-cold PBS, Fc receptors were blocked with 1 µg of anti-mouse CD16/CD32 (BioLegend, Koblenz, Germany) for 10 min at 4 °C. Next, 50 µL of a fluorochrome-conjugated antibody mix was added to detect CD4, CD8, CD11b, CD19, CD138 and TACI. All antibodies were purchased from BioLegend (BioLegend, Koblenz, Germany). Samples were incubated for further 30 min at 4 °C and immediately analyzed after washing on a flow cytometer (Beckman Coulter Gallios, Beckman Coulter GmbH, Krefeld, Germany). Kaluza analysis 2.1 software (Beckman Coulter GmbH, Krefeld, Germany) was used to analyze flow cytometric data. The median fluorescence intensity (MFI) of SNA subsets was used to determine cell surface sialylation of cell subsets. The absolute number of cell subsets was determined by multiplying the percentage of the respective subset divided by 100 with the total splenocyte number.

### 4.6. Preparation of Spleen and Paw Tissue Lysates

Tissue extracts were prepared from spleens and paws. Joints were dissected as described elsewhere [58]. Tissues were homogenized according the manufacturer’s instruction of TissueLyser LT (Qiagen, Düsseldorf, Germany). Briefly, 200 mg of tissue was applied to 500 µL modified RIPA lysis [50 mM Tris-HCl pH 7.4, 150 mM NaCl, 1 mM EDTA, 1% Triton X-100, 0.1% SDS, 1% Sodium deoxycholate supplemented with phenylmethylsulfonyl fluoride (final concentration 1 mM) and complete™ Protease Inhibitor Cocktail (Roche, Basel, Switzerland] and homogenized using steel beads for 15 min oscillating with a frequency of 50/s. Samples were centrifuged at 10.000 ×g at 4 °C for 10 min. The total protein content (mg/mL) was measured in the supernatant by BCA assay (ThermoFisher Scientific, Waltham, MA, USA).

### 4.7. Determination of Neuraminidase Activity

Neuraminidase activity was measured in the spleen and joint extracts using the NA-FluorTM Influenza Neuraminidase Assay Kit (Applied Biosystems^®^, Foster City, CA, USA). Here, the synthetic substrate 2′-(4-Methylumbelliferyl)-α-D-N-acetylneuraminic acid (4-MUNANA) is cleaved by neuraminidases resulting in fluorescent 4-Methylumbelliferone (4-MU) and N-acetylneuraminic acid (NANA). The unhydrolyzed substrate shows maxima of λEx = 315 nm/λEm = 374 nm, whereas the product 4-MU has spectral maxima at λEx = 365 nm/λEm = 450 nm. Paw and spleen extracts were diluted 1:4 in assay buffer and the assay was performed according to the manufacturer’s instructions. Fluorescence was measured over the course of 60 min using an excitation wavelength of 360 nm and an emission wavelength of 465 nm. Neuraminidase from Clostridium perfringens (Sigma Aldrich, St. Louis, MO, USA) and murine muscle lysate were used as a positive control. Activity assays were performed in duplicates. A 4-methylumbelliferone sodium salt (4-MU-SS) standard curve was generated whereby a nonlinear regression was used to calculate neuraminidase activity in tissue samples. Neuraminidase activity was normalized to the total protein content and expressed as µU/mg total protein.

### 4.8. Statistical Analysis

Statistical analysis was performed using the GraphPad Prism 8 software. Results are shown as mean value ± standard deviation (±SD) or Tukey’s box- and whisker plots. In the case of normal distribution, unpaired Student’s *t*-test (two groups) and one-way ANOVA (three groups) was used to test one parameter. Repeated measurements were analyzed with two-way ANOVA followed by post-hoc analysis. In case the normality test did not suggest normal distribution, the Mann–Whitney test was chosen. Arthritis prevalence was calculated by Fisher’s exact test. Statistical significance was defined as *p* ≤ 0.05 and rating of statistical significance was defined as * = *p* ≤ 0.05; ** = *p* ≤ 0.01; *** = *p* ≤ 0.005; **** = *p* ≤ 0.0001.

## Figures and Tables

**Figure 1 ijms-22-01428-f001:**
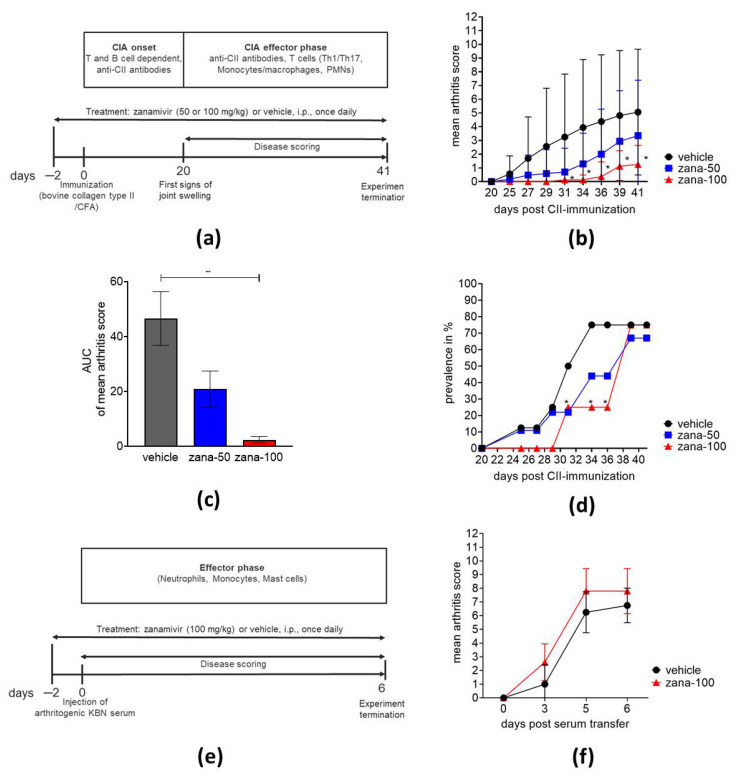
Zanamivir treatment ameliorates collagen-induced arthritis (CIA), but not serum transfer-induced arthritis (STIA). (**a**) Experimental schedule of CIA. Arthritis was induced at day 0 in DBA/1J mice by immunization with a bovine CII/CFA emulsion. Treatment was started two days before immunization. Mice received daily intraperitoneal injections of vehicle, 50 mg/kg zanamivir or 100 mg/kg zanamivir. (**b**) Severity of arthritis is expressed as mean arthritis score. (**c**) Results of “area under curve of mean arthritis score” from day 20 to day 41 are shown. ** *p* ≤ 0.01. (**d**) Prevalence (number of diseased animals (score >0) at indicated time points) is expressed in percentage (%). (**e**) Experimental schedule of STIA. Arthritis was induced at day 0 in C57BL/6J mice by KBN serum transfer. Treatment was started two days before serum transfer. Mice received daily systemic injections of vehicle or 100 mg/kg zanamivir. (**f**) Mean arthritis score of vehicle and mice treated with 100 mg/kg of zanamir in serum-transfer-induced arthritis (STIA). CIA: vehicle, *n*=16; zanamivir 50 mg/kg, *n* = 17; zanamivir 100 mg/kg, *n* = 8. STIA: vehicle, *n* = 4; zanamivir 100 mg/kg, *n* = 5. Results are shown as ± standard deviation (±SD). * *p* ≤ 0.05. For statistical analysis, we used two-way ANOVA for repeated measures with Tukey’s multiple comparisons test (**b**), Kruskal Wallis with Dunn’s post hoc test in (**c**), Fisher’s exact test (**d**), and unpaired Mann–Whitney U test in (**f**).

**Figure 2 ijms-22-01428-f002:**
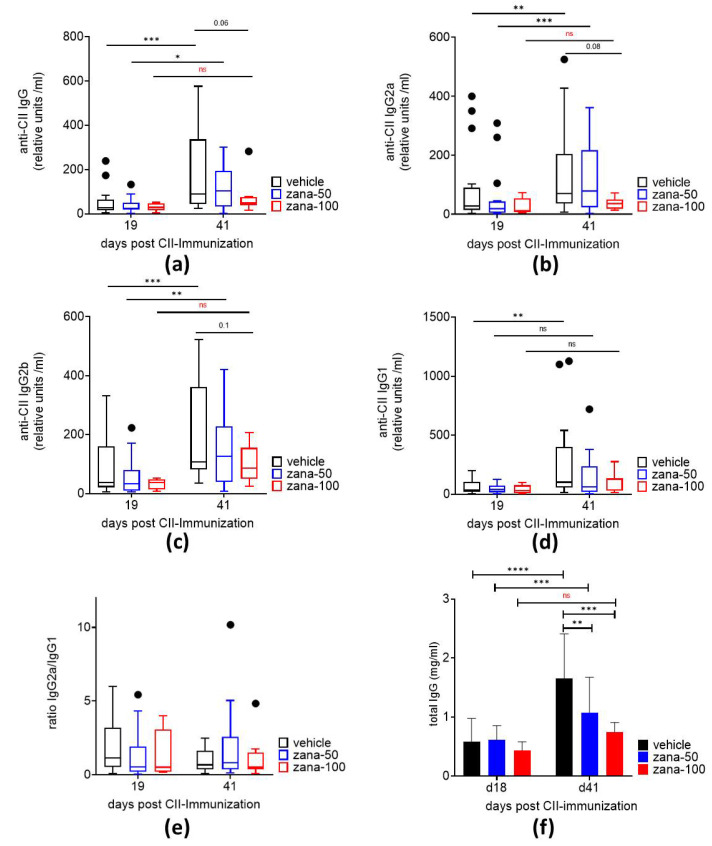
High-dose zanamivir prevents the increase in anti-CII IgG and total IgG levels in the arthritic phase of CIA. Serum levels of anti-CII IgG (**a**), anti-CII IgG2a (**b**), anti-CII IgG2b (**c**), and anti-CII IgG1 (**d**) antibodies in CIA mice. (**e**) Anti-CII IgG2a: anti-CII IgG1 ratio. Data (a-e) are presented as Tukey’s box- and whisker plots. Observations outside the whiskers are considered outliers and presented as black dots. (**f**) Serum levels of total IgG. Results are shown as ± standard deviation (±SD). * *p* ≤ 0.05; ***p* ≤ 0.01; *** *p* ≤ 0.005; **** *p* ≤ 0.0001. ns = not significant. Two-way ANOVA for repeated measures with Sidak’s or Tukey’s multiple comparisons test.

**Figure 3 ijms-22-01428-f003:**
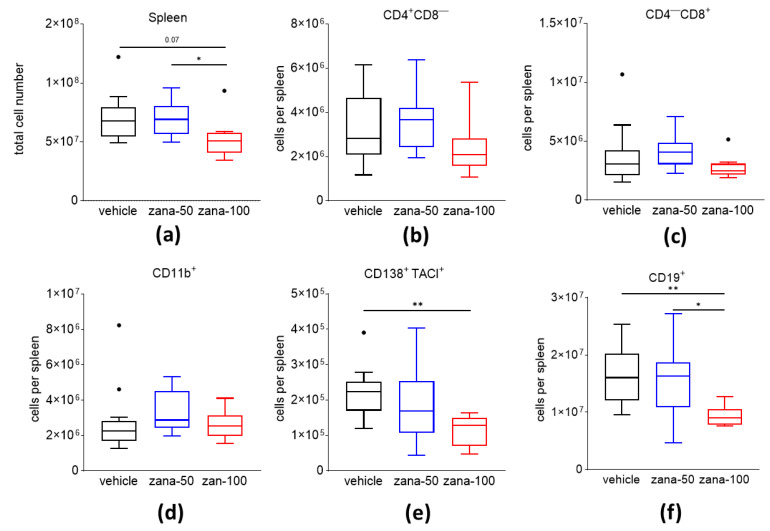
Zanamivir decreases the absolute cell number of CD138^+^/TACI^+^ plasma cells (PC) and CD19^+^ B cells in the spleens of CIA mice. (**a**) Total cell number in spleens of CIA mice. Immune cell populations (CD4^+^CD8^−^ T cells (**b**), CD4^−^/CD8^+^-T cells (**c**), CD11b^+^ cells (**d**), CD138^+^/TACI^+^ plasma cells (**e**), and CD19^+^ B cells (**f**) were analyzed by flow cytometry. Absolute cell numbers were calculated. Results are presented as Tukey’s box- and whisker plots. Observations outside the whiskers are considered outliers and presented as black dots. * *p* ≤ 0.05; ** *p* ≤ 0.01. One-way ANOVA with Tukey’s multiple comparisons test was used for statistical analysis.

**Figure 4 ijms-22-01428-f004:**
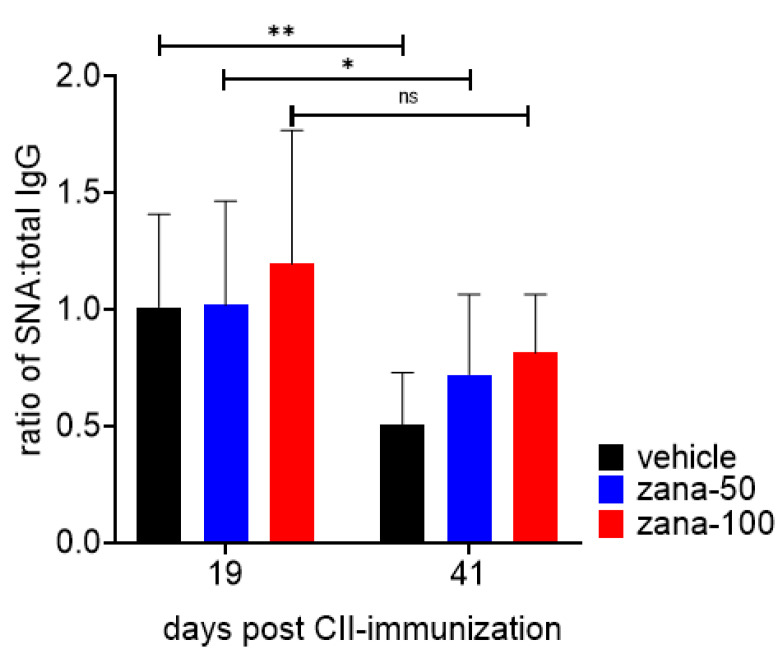
High-dose zanamivir inhibits the decrease in IgG sialylation in the arthritic phase of CIA. Levels of (2,6)-SA residues on total serum IgG were measured by ELISA using biotinylated Sambucus nigra, a lectin specific for terminal-linked (2,6)-SA. IgG sialylation is expressed as ratio of SNA binding (OD) to total IgG concentration. Results are shown as ± standard deviation (±SD). * *p* < 0.05, ** *p* < 0.01; ns = not significant. Two-way ANOVA for repeated measures with Sidak’s multiple comparisons test was used.

**Figure 5 ijms-22-01428-f005:**
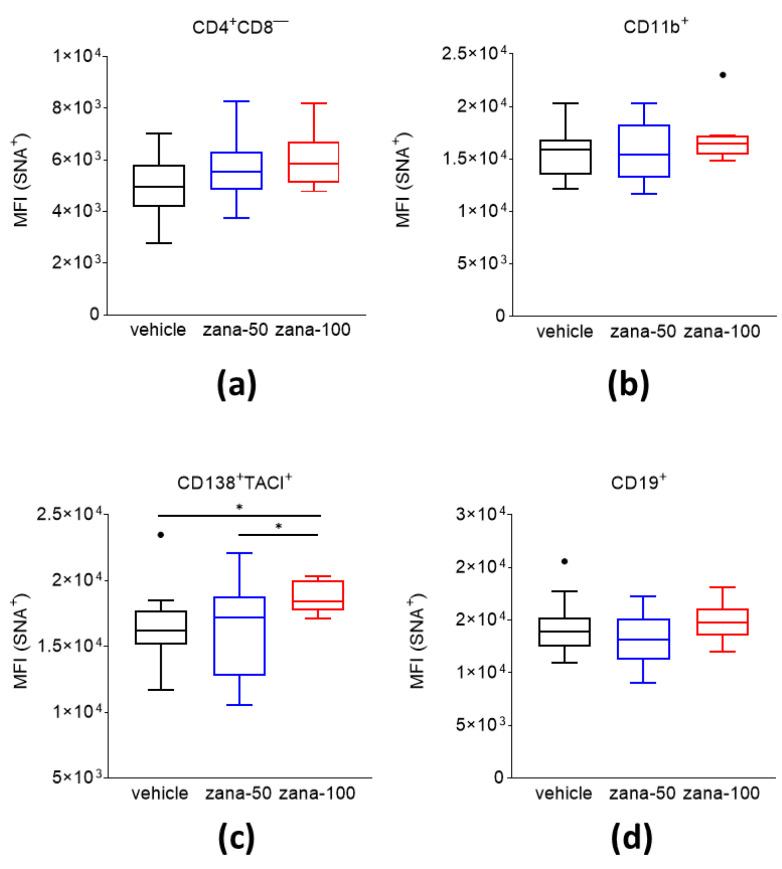
An increased level of alpha-(2,6)-sialic acids is observed on the cell surface of CD138^+^/TACI^+^ plasma cells after zanamivir treatment in CIA. Levels of (2,6)-SA residues on cell subsets—(**a**) CD4^+^CD8^−^T cells, (**b**) CD11b^+^ cells, (**c**) CD138^+^/TACI^+^ cells, and (**d**) CD19^+^cells—were measured by flow cytometric analysis. Data are presented as Tukey’s box- and whisker plots. Observations outside the whiskers are considered outliers and presented as black dots. * *p* < 0.05. Welch’s ANOVA test with Dunnett’s T3 multiple comparisons was used for statistical analysis.

**Figure 6 ijms-22-01428-f006:**
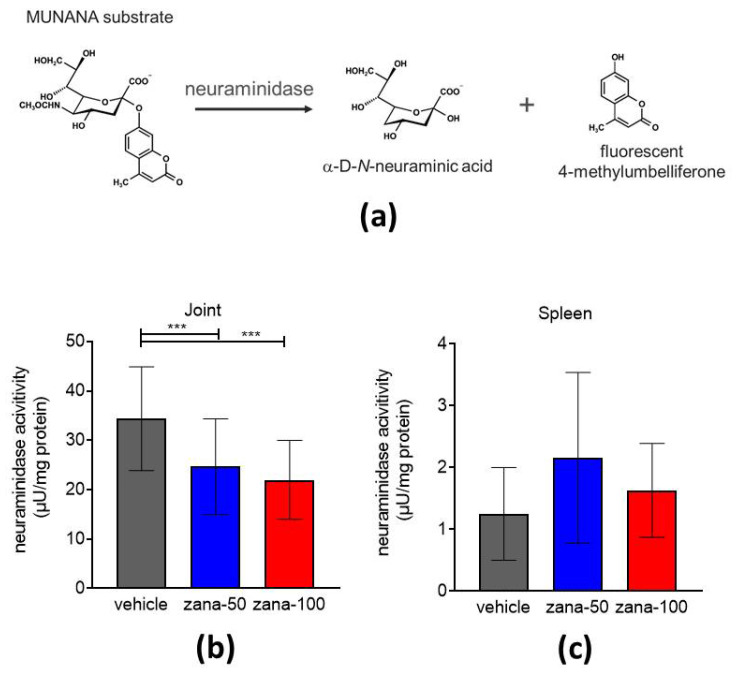
Reduced neuraminidase activity in arthritic joints upon zanamivir treatment. (**a**) 2′-(4-methylumbelliferyl)-α-D-N-acetylneuraminic acid (MUNANA) is catalyzed in the presence of neuraminidase into α-D-N-acetylneuraminic acid and fluorescent 4-methylumbelliferyl. Tissue extracts of CIA hind paws (**b**) and spleens (**c**) were prepared and applied to MUNANA assay. Neuraminidase activity was calculated as µU/mg total protein. Results are shown as ± standard deviation (±SD). *** *p* < 0.005. One-way ANOVA with Tukey’s multiple comparisons test was used for statistical analysis.

## Data Availability

The data presented in this study are available in the article.

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
