# Peer review of "Neuraminidase Inhibitor Zanamivir Ameliorates Collagen-Induced Arthritis"

_ijms, 2021, doi:10.3390/ijms22031428_

Round 1

Reviewer 1 Report

The authors shoud explain why only one injection of collagen was performed instead of two, as usually recommended. Reference 41 is incomplete.

It is stated (lines 170-172) that "mice treated with 100 mg/kg of zanamivir showed no significant elevation of anti-CII IgG, IgG2a and IgG2b levels between day 19 and day 41 indicating an inhibition of anti-CII autoantibody production". Significant effect of zanamivir should significantly decrease the observed parameters as compared to "vehicle". Please explain.
The same applies to figure 4 with regard to SNA:IgG ratio (lines 222 and 223). 
In this regard, the titles of figures 2 and 4 are misleading.

Lines 266-268: Rephrase both sentences "In joints, vehicle-treated mice showed significantly higher amounts of neuraminidase activity compared to zanamivir-treated CIA mice (Figure 6B). Overall, splenocytes of CIA mice had a >10 fold lower neuraminidase activity in splenocytes than in joints.

Lines 300-301: "We showed that zanamivir at 100 mg/kg significantly inhibits the production of anti-CII IgG antibodies and total IgG concentrations during CIA" is not accurate.

Reviewer 2 Report

This is an original research evaluating, for the first time, the therapeutic potential of neuramidases (NEUs) activity inhibition in rheumatoid arthritis, using Zanamivir in mice models of the disease (CIA and STIA). The presented results pointed out NEUs as new molecular targets for RA management, thus prompting future research on the field.

I have the following comments to improve the presentation and clarify some issues:

Line 38: Other treatments besides anti TNF-alpha therapy are currently used in RA management (as mentioned in discussion). Please rephrase to briefly mention them and their limitations.

Line 94-101: This statement describes the results of the work. Please consider to remove it from Introduction.

Line 117: Why treatment started 2 days before C-II immunization? What is the effect of Zana after CIA immunization? Were control mice injected with complete Freund’s adjuvant (CFA) used? Were the potential adverse or toxic effects of zana treatment evaluated?

Line 118: Although two independent observers under blinded conditions graded arthritis score, conclusions on zanamivir treatment on CIA clinical amelioration are based only in this observer-dependent scale. Of note, high deviations of mean arthritis score are observed (figure 1b). Thus, other parameters should also be used, such as paw thickness and histopathology analysis to evaluate joint architecture (synovitis, cartilage loss and bone erosions).

Line 337: Could the effect of NEUs inhibitors be counterbalanced by an increase of STs activity? Please discuss.

Round 2

Reviewer 2 Report

Thanks to the authors for preparing a revised version of your manuscript properly addressing all of my comments. However, a minor revision is still needed:

Line 127: "effective neuramidase concentrations" or "effective neuramidase activity inhibition"???

Author Response

Minor revision

Response to Reviewer 2 Comments

Comments and Suggestions for Authors

Thanks to the authors for preparing a revised version of your manuscript properly addressing all of my comments. However, a minor revision is still needed:

Line 127: "effective neuramidase concentrations" or "effective neuramidase activity inhibition"???

Reponse:

We would like to thank the reviewer for his important comment.

We corrected the text in line 116 (former line 127) as followed: …” Treatment started two days before CII-immunization to ensure effective inhibition of neuraminidase activity at the time of immunization.”